# Study on Creep-Fatigue Mechanical Behavior and Life Prediction of Ti_2_AlNb-Based Alloy

**DOI:** 10.3390/ma15186238

**Published:** 2022-09-08

**Authors:** Yanju Wang, Xinhao Wang, Yanfeng Yang, Xiang Lan, Zhao Zhang, Heng Li

**Affiliations:** 1Materials Evaluation Center for Aeronautical and Aeroengine Application, AECC Beijing Institute of Aeronautical Materials, Beijing 100095, China; 2State Key Laboratory of Solidification Processing, Northwestern Polytechnical University, Xi’an 710072, China

**Keywords:** Ti_2_AlNb-based alloy, creep-fatigue, fracture mechanism, life prediction

## Abstract

Low-cycle fatigue, creep and creep-fatigue tests of Ti_2_AlNb-based alloy were carried out at 550 °C. Compared with low-cycle fatigue, a creep-fatigue hysteresis loop has larger area and smaller average stress. The introduction of creep damage will greatly reduce the cycle life, and change the fatigue crack initiation point and failure mechanism. Based on the linear damage accumulation rule, the fatigue damage and creep damage were described by the life fraction method and the time fraction method, respectively, and the creep-fatigue life of the Ti_2_AlNb-based alloy is predicted within an error band of ±2 times.

## 1. Introduction

Creep-fatigue failure may occur in the high-temperature structures of aeroengines during operation, which is the result of power changes during equipment operation. Power transience will lead to cyclic loads, and stress relaxation will occur during stable operation [1]. Creep failure is usually caused by creep voids near grain boundaries, and fatigue failure is usually caused by crack damage. In creep-fatigue, the interaction between the two damages will lead to accelerated failure and shorten the service life of components [2]. Creep-fatigue interaction poses a challenge to life prediction of high-temperature components. In particular, the application of new materials increases the uncertainty of the creep-fatigue interaction and its life prediction. However, the existing creep-fatigue life prediction models may not be suitable for new materials [3]. Ti_2_AlNb-based alloy is an intermetallic compound. Its long-range ordered super-lattice structure has the effect of weakening dislocation diffusion and high-temperature diffusion, which makes the alloy have the advantages of high specific strength, specific stiffness, creep resistance, fracture toughness and excellent oxidation resistance [4,5]. As a new generation of aerospace lightweight high-temperature structural materials, Ti_2_AlNb-based alloy is expected to partially replace Ni-based alloy for the manufacture of aeroengine structural parts to reduce the weight of the engine [6,7]. At present, creep-fatigue related research mainly focuses on nickel-based alloy, 316 stainless steel and P91 steel and other traditional high- temperature structural materials [8,9,10,11,12]. There are few public reports on creep-fatigue studies of Ti_2_AlNb-based alloys. Therefore, it is urgently needed to study the creep-fatigue behavior of Ti_2_AlNb-based alloy and evaluate its creep-fatigue life.

The creep-fatigue test sets a stress or strain holding in the fatigue cycle to introduce creep damage and explore the interaction between fatigue damage and creep damage. According to the different holding types in the cycle, creep-fatigue can be divided into tensile-holding creep-fatigue, compression-holding creep-fatigue and tensile-compression-holding creep-fatigue. Relevant experimental studies show that the cycle life will be reduced after the introduction of creep damage, and for most materials, tensile loading is more harmful than compressive loading [2,13,14,15,16]. Most of the creep-fatigue tests are strain loading. Because the creep-fatigue peak stress is often higher than the yield point of the material, stress fluctuations will cause large strain fluctuations, and strain loading can reduce this error. In the study, low-cycle fatigue and creep-fatigue were both under strain loading.

A series of creep-fatigue damage calculation methods are introduced in standards including ASME III [17], R5 [18] and RCC-MR [19]. Most of these international standards are based on the linear damage accumulation law to predict creep fatigue life. The creep-fatigue life prediction model based on the linear damage accumulation law is easy to operate, has fewer material parameters and is widely used [20,21]. In this method, the fatigue damage and creep damage in the creep-fatigue process of materials are calculated, and then they are linearly added to represent the creep-fatigue damage. When the creep-fatigue damage reaches the critical value, creep-fatigue failure occurs. Usually [3], the creep-fatigue steady-state hysteresis loop data are used to calculate the weekly damage. The critical value of damage is set to 1, and the cycle life is the reciprocal of the weekly damage.

In this paper, the purpose is to study the creep-fatigue interaction of Ti_2_AlNb-based alloy and predict its creep-fatigue life. Firstly, low-cycle fatigue, creep and creep-fatigue tests of Ti_2_AlNb-based alloy were carried out at 550 °C to analyze the effect of creep damage on the cyclic deformation and damage accumulation of materials. Then, based on the linear cumulative damage criterion, a life prediction model was established to predict the creep-fatigue life of Ti_2_AlNb-based alloy.

## 2. Experimental Materials and Design

The combustion chamber casing is an annular thin-walled structure. The casing blank was obtained by forging, and then the finished product was obtained by machining the blank. In this paper, Ti_2_AlNb-based alloy is cut from the combustion chamber casing blank. The microstructure of the initial Ti_2_AlNb-based alloy was characterized at room temperature, and the results are shown in Figure 1. EBSD and EDS analyses were performed using a ZEISS Sigma 300 scanning electron microscope, which was equipped with an electron backscatterer and an energy spectrometer. It can be seen from the metallographic photograph that the grain size of the alloy is between 100 um and 500 um. The black acicular O phase is distributed in the grey β phase. It can be seen that the distribution of the O phase in different grains is not uniform. The black spots in Figure 1a are defects in the preparation process of metallographic samples, not the microstructure characteristics of the material. According to the results of EBSD analysis, Ti_2_AlNb-based alloy is composed of the O phase, the β phase and the $a_{2}$ phase, accounting for 52%, 30% and 7% of the area, respectively, and the unresolved rate is 11%. EDS analysis results show that the main components of the alloy are Ti, Al, Nb and Zr, and the atomic content ratio is 55:21:23:1.

Low-cycle fatigue, creep and creep-fatigue tests were carried out at 550 °C. The comparative analysis of three experimental results is helpful to study the creep-fatigue interaction of Ti_2_AlNb-based alloy. In addition, low-cycle fatigue and creep tests provide necessary data support for parameter fitting of the creep-fatigue life prediction model. Figure 2 shows the sample sizes of different tests. Low-cycle fatigue tests and creep-fatigue tests were carried out on the Instron-8802 fatigue testing machine. When the sample fracture or the stress of the testing machine decreased by 25% instantaneously, the failure of the sample was judged. As shown in Figure 3, the loading waveforms of low-cycle fatigue and creep-fatigue tests are triangular wave and trapezoidal wave, respectively, and the strain ratio is −1. Creep tests were carried out on a GNCJ-100E creep testing machine. The equipment, manufactured by GangYanNaKe Testing Technology Co., LTD., can perform creep tests of 1~100 kN. When the sample was broken, the failure of the sample was judged. Table 1 gives the specific test parameters. The sample processing and tests were carried out according to GB/T 38822-2020, GB/T 15248-2008 and GB/T 2039-2012.

## 3. Creep-Fatigue Life Prediction Modeling

In 1945, Miner [22] used the ‘life fraction’ method to calculate fatigue damage per cycle. After that, Robinson [23] used the ‘time fraction’ method to calculate the creep damage of the material in the process of studying the high-temperature creep fracture behavior of the alloy. Taira [24] used the above methods to calculate the fatigue damage and creep damage in the creep-fatigue process, and then the two kinds of damage were linearly superimposed to represent the creep-fatigue damage. The equation for the time fraction approach is as follows:(1)$$\sum_{i}^{N}\left[\frac{1}{N_{f}\left(T,\Delta \varepsilon \right)}+\frac{t_{i}}{t_{r}\left(T,\sigma \right)}\right]=1$$

In the formula, N is the number of cycles, $N_{f}\left(T,\Delta \varepsilon \right)$ is the fatigue life related to the test temperature and strain range, $t_{i}$ is the holding time, and $t_{r}\left(T,\sigma \right)$ is the creep life related to the test temperature and stress level. The front term of Equation (1) is the fatigue damage of each cycle, which can be expressed as:(2)$$d_{f}=\frac{1}{N_{f}(T,\Delta \varepsilon )}$$

In the formula, $d_{f}$ is fatigue damage per cycle. The latter term of Equation (1) is creep damage per cycle. However, stress relaxation occurs during the creep-fatigue strain loading stage, and the tensile stress of the specimen decreases continuously. The $t_{r}$ in Equation (1) is usually replaced by the creep life of the material at the same test temperature and the initial stress level of the holding. This will overestimate the creep damage per cycle. To describe the creep damage accumulated in the holding stage more reasonably, it is changed into the integral form:(3)$$d_{c}=\int_{0}^{t_{h}}\frac{dt}{t_{r}(T,\sigma )}$$

In the formula, $d_{c}$ is the creep damage per cycle calculated by the time fraction method and $t_{h}$ is the holding time.

The creep-fatigue cycle life can be expressed as follows:(4)$$N_{c-f}=1/\left(d_{f}+d_{c}\right)$$

According to Equations (2) and (3), it is necessary to fit the low-cycle fatigue and creep life and describe the stress drop process in the holding stage. The fitting formula is as follows [3,25]:(5)$$N_{f}{\left(\Delta \varepsilon -\Delta \varepsilon_{0}\right)}^{m}=c$$
(6)$$\sigma =a\log\left(t_{r}\right)+b$$
(7)$$\sigma \left(t\right)=\sigma_{0}-K\log\left(1+t\right)$$

In the formula, $\Delta \varepsilon_{0}$, m, c, a, b and K are material constants and $\sigma_{0}$ is the stress at the beginning of holding. The creep-fatigue life prediction model of Ti_2_AlNb- based alloy can be obtained by combining Equations (2)–(7):(8)$$N_{c-f}=1/\left\{\frac{{\left(\Delta \varepsilon -\Delta \varepsilon_{0}\right)}^{m}}{c}+\exp\left(\frac{b-\sigma_{0}}{a}\right)\frac{a}{K+a}\left[{\left(1+t_{h}\right)}^{\frac{K+a}{a}}-1\right]\right\}$$

## 4. Results and Discussion

### 4.1. Creep-Fatigue Behavior

In the process of low-cycle fatigue and creep-fatigue cyclic deformation, the stress-strain curve of metal materials will gradually become a closed ‘loop’, that is, the hysteresis loop. The hysteresis loop is an intuitive reflection of the mechanical behavior in the process of material cyclic deformation. With the increase in cycle times, the hysteresis loop tends to be stable and almost no longer changes, which is called a steady-state hysteresis loop. Steady-state hysteresis loops are one of the basic data reflecting fatigue cyclic deformation behavior. Generally, the hysteresis loop of metal materials tends to be stable after 1/3 of the fatigue life, so the half-life hysteresis loop is often regarded as a steady-state hysteresis loop in engineering. Figure 4a shows the steady-state hysteresis loop of low-cycle fatigue of Ti_2_AlNb-based alloy. It can be seen that the curve gradually widens with the increase in strain range. The area surrounded by the hysteresis loop represents the difference between the absorbed energy at loading and the released energy at unloading, which can be used to characterize the dissipation of strain energy. The larger the area enclosed by the hysteresis loop, the more fatigue damage accumulated per cycle. Figure 4b shows the steady-state hysteresis loop of creep-fatigue of Ti_2_AlNb-based alloy. Stress relaxation occurs during the holding period. With the increase in holding time, the area surrounded by the hysteresis loop increases, and the hysteresis loop moves downward. Low-cycle fatigue and creep-fatigue steady-state hysteresis loops have no obvious boundary between linear and nonlinear segments. This is caused by the Bauschinger effect in the continuous loading–unloading–reverse loading process, and the elastic limit decreases continuously under reverse loading.

The concept of normalized life is introduced, that is, the ratio of current cycle number to the cycle life (N/N_f_ and N/N_c–f_). The cyclic soft/hardening behavior is characterized by the evolution of the maximum tensile stress of each cycle with the normalized life. It can be seen in Figure 5a that the specimens at different strain levels show slight cyclic hardening in the front and middle stages of low-cycle fatigue, which is caused by strain hardening during cyclic deformation. At the end of the curve, there is obvious cyclic softening, because the initiation and propagation of fatigue cracks reduce the effective bearing area, resulting in the decrease in tensile stress. It can be seen from Figure 5b that there is a significant softening at the beginning of creep-fatigue and no rapid softening at the end of the curve. This indicates that under creep-fatigue load, the crack expands rapidly after initiation and causes the failure of the specimen in one cycle.

Figure 6 shows the creep curve of Ti_2_AlNb-based alloy. At 400~600 MPa creep stress levels, there is only an initial creep stage and a steady creep stage before creep rupture, and the steady creep rate increases with the increase in creep stress level.

Table 2 shows the low-cycle fatigue, creep and creep-fatigue life of Ti_2_AlNb-based alloy at 550 °C. It can be found that the low-cycle fatigue and creep life generally decrease with the increase in strain range and stress level. But creep-fatigue life does not monotonically decrease with the increase in holding time. It can be seen from Figure 7 that after creep damage is introduced into low-cycle fatigue, the cycle life decreases significantly. When the holding time increases from 50 s to 100 s, the cycle life increases. In addition, when the holding time increases from 50 s to 100 s, the average stress of the steady-state hysteresis loop decreases significantly, which leads to the increase in life. The creep-fatigue life of Ti_2_AlNb-based alloy is affected by the holding time and average stress. The larger the holding time and average stress, the lower the creep-fatigue life of the sample.

### 4.2. Creep-Fatigue Fracture

Low-cycle fatigue, creep and creep-fatigue tests are carried out for a long time at high temperature, and it is difficult to observe the changes in the internal structure of the specimen during the tests. However, the information of the damage initiation and accumulation process will be left on the fracture surface [26]. The fracture of Ti_2_AlNb-based alloy specimens is observed to analyze the failure mode. By comparing the failure modes under different loading conditions, the effect of creep damage on fatigue failure of Ti_2_AlNb-based alloy is investigated.

The fatigue fracture observation results are shown in Figure 8. In the process of low-cycle fatigue, crack initiation and propagation account for most of the time of fatigue life. Therefore, this paper will focus on the comparison of crack initiation and propagation regions under different strain levels, which has been marked by red frames in the diagram. It can be seen that when the strain level is low, the fracture mode of crack initiation and propagation zone is transgranular fracture, and when the strain level increases, the low-cycle fatigue fracture mode of Ti_2_AlNb-based alloy changes from transgranular fracture to transgranular and intergranular mixed fracture.

The creep fracture is shown in Figure 9. It can be seen that the creep section of Ti_2_AlNb-based alloy consists of an intergranular fracture zone and a transgranular fracture zone. Affected by creep damage, voids are formed near the grain boundary of Ti_2_AlNb-based alloy, and the voids grow up to form intergranular cracks. The growth of creep cavities and the propagation of intergranular cracks lead to the decrease of the effective bearing area of the sample, and the actual stress on the effective bearing surface will continue to increase, resulting in the crack. In addition, it can be seen in Figure 9 that there are a large number of secondary cracks between the intergranular fracture zone and the transgranular fracture zone of the fracture surface. This part is the interface between the cracked zone and the uncracked zone when the specimen is about to undergo creep failure. Severe geometric discontinuity leads to stress concentration and promotes secondary crack propagation.

The creep-fatigue fracture of Ti_2_AlNb-based alloy is shown in Figure 10. The creep-fatigue test fracture with 50~100 s holding time is composed of small planes in different directions, which is a typical intergranular brittle fracture. When the holding time increases to 150~300 s, the fracture is similar to a creep fracture. Figure 11 shows the fracture comparison of fatigue, creep, and creep-fatigue tests. After creep damage was introduced into the fatigue test of Ti_2_AlNb based alloy, the fracture morphology obviously changed. The fatigue crack of Ti_2_AlNb-based alloy initiates at the edge of the sample, and the final fracture is a transgranular and intergranular mixed fracture. After adding strain holding, the failure mode in the early stage of the test changed into intergranular fracture. The transgranular fracture zone in Figure 10c,d is formed at the moment of specimen fracture. Most importantly, the crack source can no longer be observed at the fracture edge. The introduction of creep damage changes the location of fatigue crack initiation. Based on the above experimental phenomena, the creep-fatigue failure process of Ti_2_AlNb-based alloy is judged as follows: creep damage leads to voids near the grain boundary, resulting in stress concentration, and fatigue cracks initiate near the grain boundary. Guided by the stress concentration at the grain boundary, the cracks propagate along the grain boundary, resulting in intergranular fracture. When the holding time increases to more than 150 s, the reason for the occurrence of the tearing fracture zone is the same as that in the creep test, which is caused by the decrease in effective bearing surface during the holding process.

### 4.3. Creep-Fatigue Life Prediction

The model parameters needed for fitting in the time-fraction creep-fatigue life prediction model are shown in Table 3. The fatigue damage parameters and creep damage parameters were fitted based on the low-cycle fatigue and creep test data of Ti_2_AlNb-based alloy, and the stress relaxation parameters were fitted based on the creep-fatigue steady-state hysteresis loop data. The fitting results are shown in Figure 12. The above parameter fitting process was carried out in Matlab.

Table 4 shows the stress at the beginning of creep-fatigue steady-state hysteresis loop. The above parameters and data were put into the model to predict creep-fatigue life, as shown in Figure 13. It can be seen that all data points fall within the error band of ±2 times, indicating that the time fraction method can well predict the creep-fatigue life of Ti_2_AlNb-based alloy.

## 5. Conclusions

In this paper, the low-cycle fatigue, creep and creep-fatigue tests of Ti_2_AlNb-based alloy at 550 °C are systematically designed, and the creep-fatigue mechanical behavior and damage mechanism of Ti_2_AlNb based alloy are explored. On this basis, the creep-fatigue life prediction model of Ti_2_AlNb-based alloy is established, which provides technical support for the creep-fatigue service performance evaluation of Ti_2_AlNb-based alloy. The paper is summarized as follows:

(1) Ti_2_AlNb-based alloy exhibits slight cyclic hardening during the low-cycle fatigue test. Since the load holding is increased at the maximum tensile strain, the maximum tensile stress of creep-fatigue decreases with the cycle times, and the maximum compressive stress increases, showing that the hysteresis loop moves downward with the cycle times. The Bauschinger effect occurs in Ti_2_AlNb-based alloy during low cycle fatigue and creep-fatigue cyclic deformation, resulting in no obvious boundary between elastic and inelastic segments in the steady-state hysteresis loop.

(2) Compared with low-cycle fatigue, the creep-fatigue life of Ti_2_AlNb based alloy is significantly reduced. When the strain range is constant, the creep-fatigue life is mainly affected by holding time and average stress. The increase in holding time and average stress will lead to the decrease in creep-fatigue life.

(3) During the creep-fatigue process of Ti_2_AlNb-based alloy, creep damage leads to voids near the grain boundary, and causes stress concentration. Cracks initiate near the grain boundary and propagate along the grain boundary, resulting in fracture of the specimen.

(4) Based on the linear damage accumulation theory, the creep-fatigue life of Ti_2_AlNb based alloy was predicted by the time fraction method, and the predicted results were within ±2 times error band.

## Figures and Tables

**Figure 1 materials-15-06238-f001:**
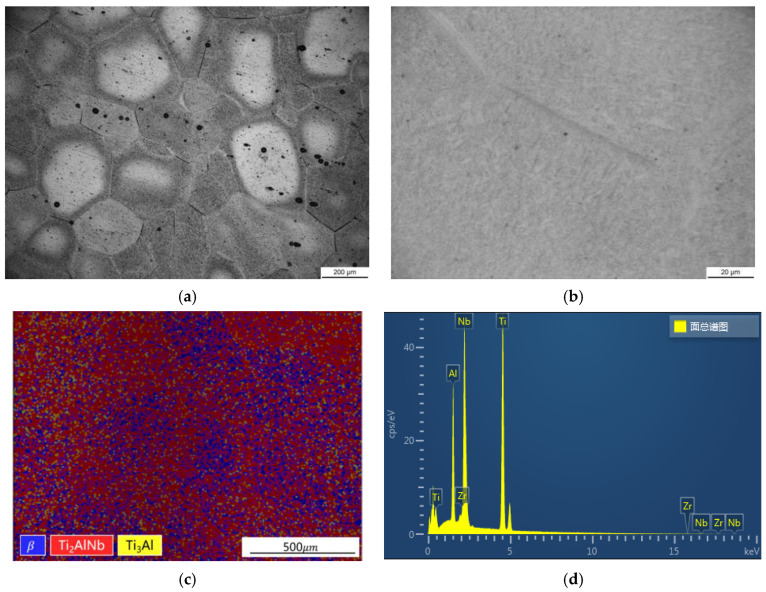
Microstructure characterization and element proportion of Ti_2_AlNb-base alloy: (**a**) OM (200×); (**b**) OM (1000×); (**c**) EBSD; (**d**) EDS.

**Figure 2 materials-15-06238-f002:**
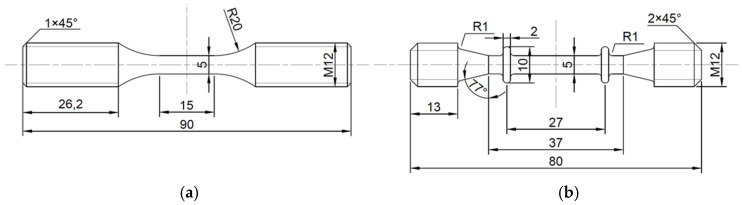
Shape and dimensions of the tested specimens: (**a**) low-cycle fatigue and creep-fatigue; (**b**) creep.

**Figure 3 materials-15-06238-f003:**
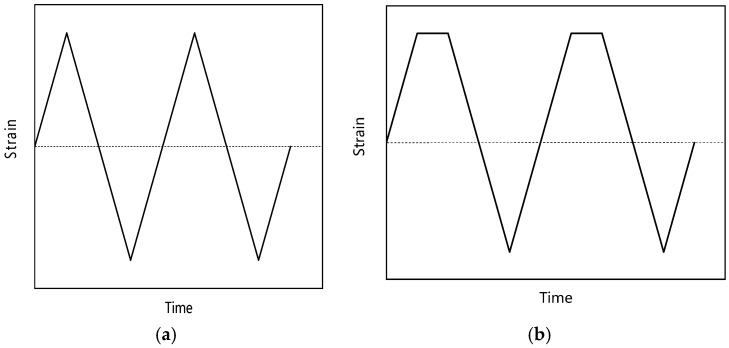
Loading waveform: (**a**) fatigue; (**b**) creep-fatigue.

**Figure 4 materials-15-06238-f004:**
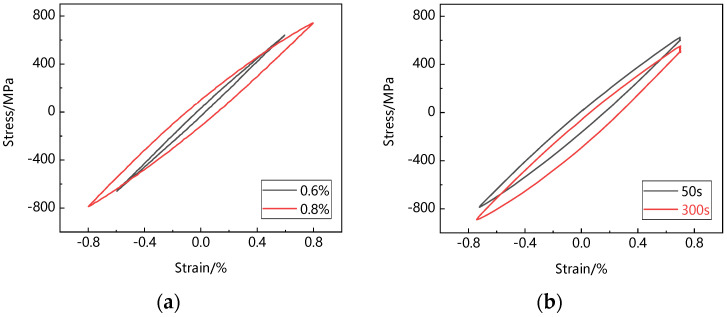
Steady-state hysteresis loop: (**a**) low-cycle fatigue; (**b**) creep-fatigue.

**Figure 5 materials-15-06238-f005:**
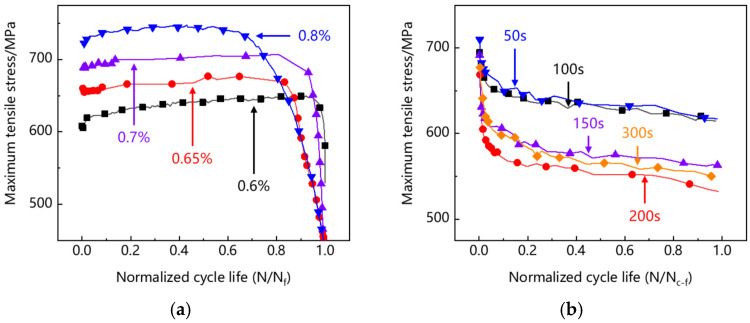
Evolution of maximum tensile stress per cycle: (**a**) low-cycle fatigue; (**b**) creep-fatigue.

**Figure 6 materials-15-06238-f006:**
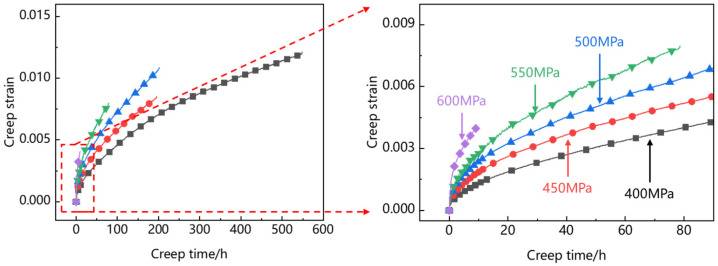
Creep curve.

**Figure 7 materials-15-06238-f007:**
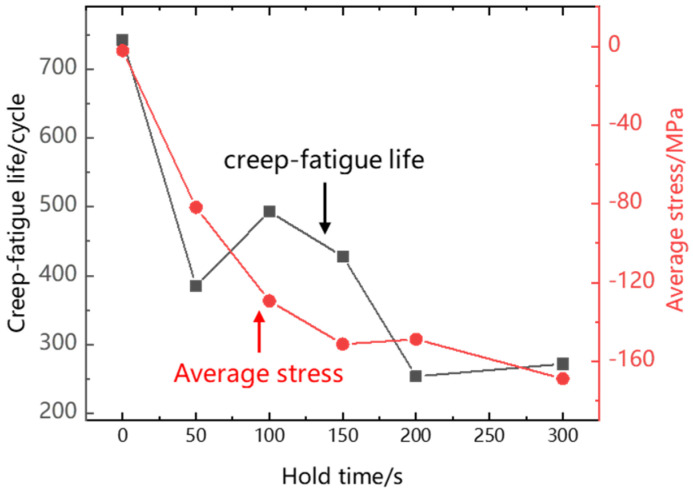
Creep-fatigue life and average stress.

**Figure 8 materials-15-06238-f008:**
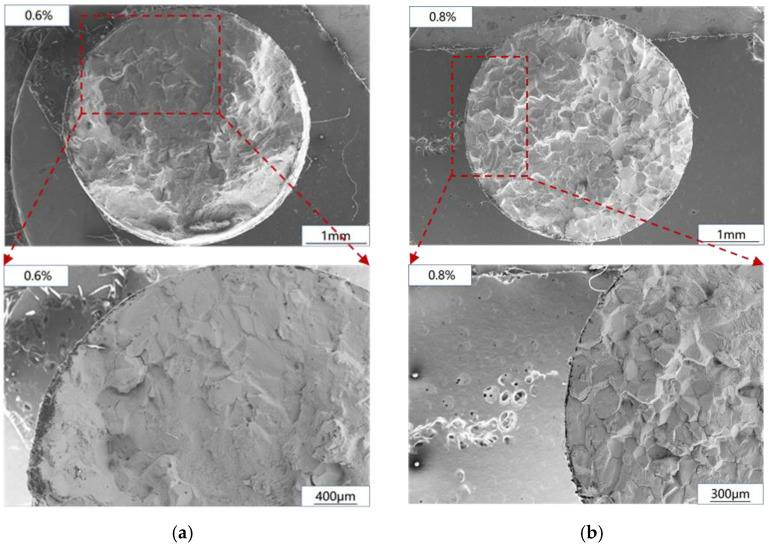
Low-cycle fatigue fracture: (**a**) 0.6%; (**b**) 0.8%.

**Figure 9 materials-15-06238-f009:**
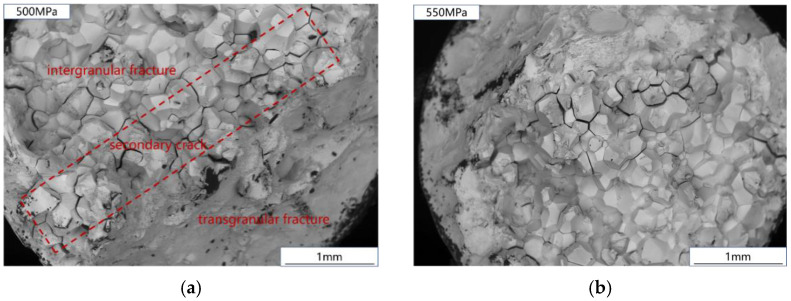
Creep fracture: (**a**) 500 MPa; (**b**) 550 MPa.

**Figure 10 materials-15-06238-f010:**
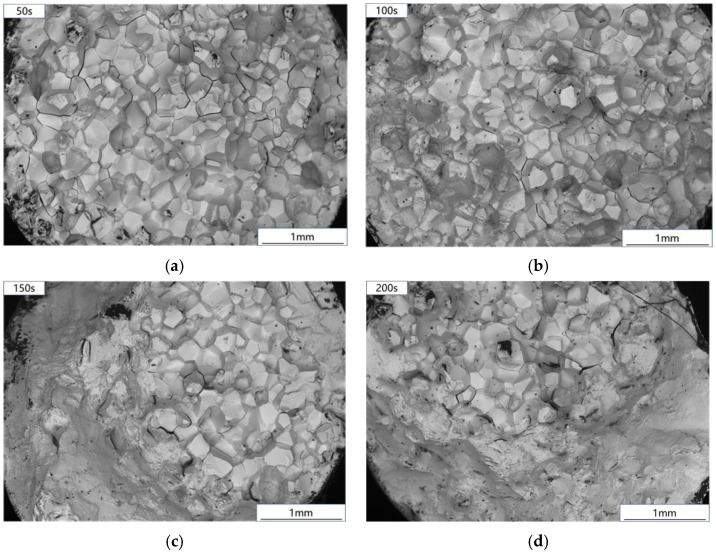
Creep-fatigue fracture: (**a**) 50 s; (**b**) 100 s; (**c**) 150 s; (**d**) 200 s.

**Figure 11 materials-15-06238-f011:**
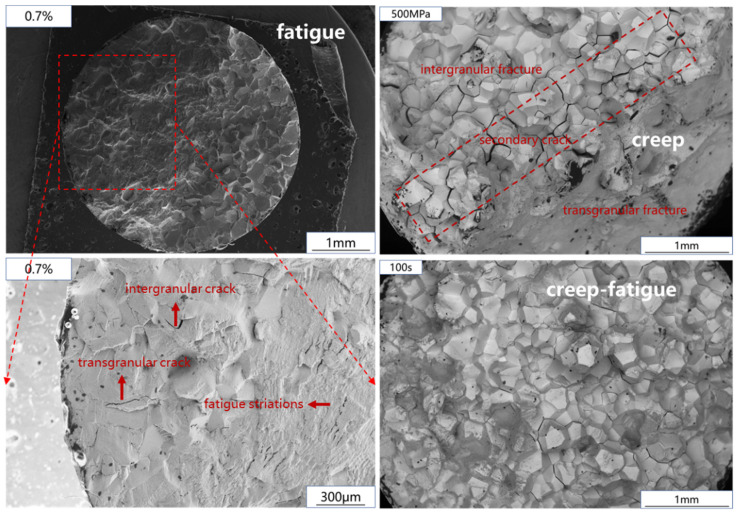
Fracture comparison of fatigue, creep and creep-fatigue.

**Figure 12 materials-15-06238-f012:**
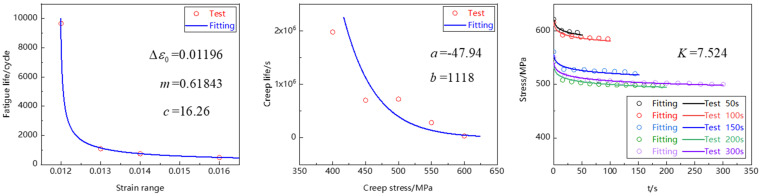
Results of parameter fitting of creep-fatigue life prediction model.

**Figure 13 materials-15-06238-f013:**
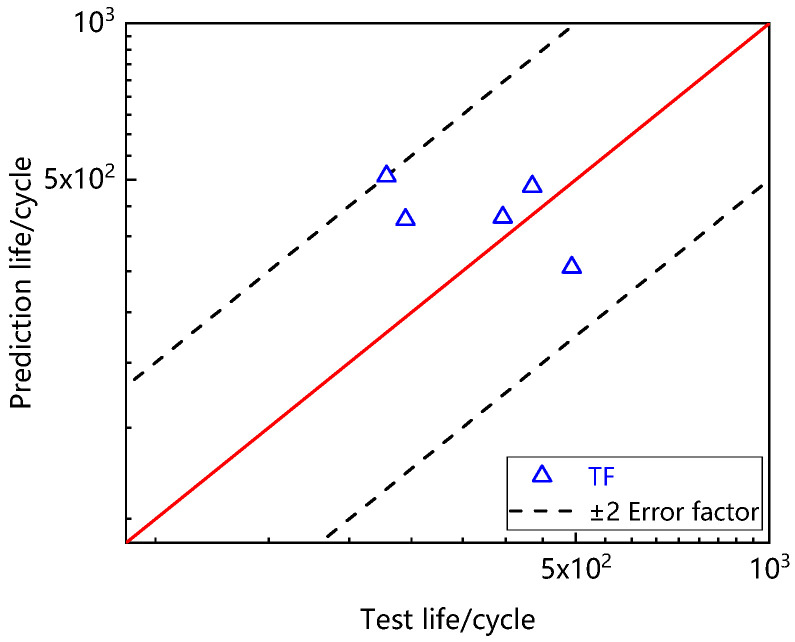
Prediction result for creep-fatigue life by time fraction method.

**Table 1 materials-15-06238-t001:** Test parameters.

Test	Loading Rate/s^−1^	Strain Range/%	Stress Level/MPa	Hold Time/s
Fatigue	0.005	1.2	\	\
		1.3		
		1.4		
		1.6		
Creep	\	\	400	\
			450	
			500	
			550	
			600	
Creep-fatigue	0.005	1.4	\	50
				100
				150
				200
				300

**Table 2 materials-15-06238-t002:** Fatigue, creep and creep-fatigue life.

Test	Strain Range/%	Stress Level/MPa	Hold Time/s	Life
Fatigue	1.2	\	\	9660/cycle
	1.3			1081/cycle
	1.4			742/cycle
	1.6			482/cycle
Creep	\	400	\	548/h
		450		195/h
		500		201/h
		550		78/h
		600		9/h
Creep-fatigue	1.4	\	50	385/cycle
			100	493/cycle
			150	428/cycle
			200	254/cycle
			300	272/cycle

**Table 3 materials-15-06238-t003:** Creep-fatigue life prediction model parameters.

Fatigue Damage	Creep Damage	Stress Relaxation
$\Delta \varepsilon_{0},m,c$	a,b	K

**Table 4 materials-15-06238-t004:** $\sigma_{0}$ of creep-fatigue steady-state hysteresis loop under different holding times.

$t_{h}$/s	50	100	150	200	300
$\sigma_{0}$/MPa	621.890	615.718	560.482	542.131	550.315

## Data Availability

Due to the lack of unanimous consent from all the authors. Data is not shared.

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
