# Peer review of "Study on Creep-Fatigue Mechanical Behavior and Life Prediction of Ti2AlNb-Based Alloy"

_materials, 2022, doi:10.3390/ma15186238_

Round 1

Reviewer 1 Report

The authors investigated the fatigue, creep and creep-fatigue behaviours of the Ti2AlNb-based alloy. I will say congratulations to the authors for the succinctly written article. However, the manuscript can be further improved by addressing the following comments:

-          Information about the source of the Ti2AlNb-based alloy will be necessary. Is it a cast material?

-          The black spots in Figure 1a are what? Information on this is important.

-          Kindly label the phases and essential features in Figure 1a

-          The low-cycle fatigue, creep and creep-fatigue tests of the Ti2AlNb-based alloy were carried out at 550 °C. Clearly state whether the microstructure presented in Figure 1 is for the as-received material or after subjection to heating at 550 °C.  

-          Please, provide clear description on how the test samples (fatigue, and creep) were produced.

-          Subscripts should be clearly written. For instance, see “The tr in Eqs. (2) …”, the subscript is not correctly written

-          Can the authors provide elucidations on the impact of the strengthening phase of the Ti2AlNb-based alloy on the fatigue failure or creep and creep-fatigue failures of the alloy?

-          Most of the references cited are old. Please, kindly update the references of this manuscript

Reviewer 2 Report

The manuscript entitled “Study on creep- fatigue mechanical behavior and life prediction of Ti2AlNb-based alloy” bears a good effort in investigating the creep-fatigue life of Ti2AlNb-based alloy but the authors have failed to represent the paper in an interesting way. The presented manuscript consists of a few grammatical errors. 

However, the manuscript needs revision to the following points. 

1)        Authors are requested to provide the detailed procedure of the material development.

2)        Authors should provide a chemical composition of the developed alloy in table format.

3)        Author must provide the make, origin and capacity of the instruments used for characterization or experiments like EBSD, EDS, creep test etc.

4)        Authors must provide the standards followed during sample preparation of creep, low cycle creep fatigue etc,

5)        Page No. 2, Line 70-71, “Ti2AlNb-based alloy is composed of O phase, β Phase and α2 phase”. Authors are requested to provide the compositions of these phases.

6)        Page No. 2 Line 67, “The microstructure of Ti2AlNb-based alloy….” The authors are requested to provide the details from where the microstructures were obtained. Also, state that Ti2AlNb-based alloy was directly taken from somewhere or self-developed?

7)        Authors are requested to explain the mechanism behind the dispersing of the acicular O phase in the β phase.

8)        It is unclear that what the authors want to state in Figure 1(a and b). If it is a general fact then it should to represent in a different way and if it is a result as stated by the authors then it should be stated separately in the result and discussion part.

9)        Author must provide the values of lattice constants along with the structures of the phases present.

10)    Authors should provide the dimensional scale for figure 2.

11)    Authors have a few bad sentence constructions in the Introduction section. Like, Line 20-21, Line 28-29 etc. Authors are requested to have a careful study of the Introduction part and modify the grammatical errors, and bad sentence constructions accordingly.

12)    This manuscript has lack of referencing. Authors are requested to provide references where required Like, Line34-35, Line36, Line 37, Line 46, Line 51-52, Line 53, 57-58, etc.

13)     Page 2, Line 47-48, Authors must state which error is talked about in “strain loading can reduce this error”

14)    Page 2, Line 69, “grain size of the alloy is between 100…..” How do the authors predict the sizes?

15)     Page 5, Line 112, “The tr in Eqs. (2)”, Where is “tr” in Equation 2?

16)     What is df?

17)    Authors are requested to explain Figure 12 broadly on how they obtained the values.

Reviewer 3 Report

This paper investigates fatigue, creep-fatigue, and creep properties for Ti2AlNb systematically investigated. Using the experimental results, prediction of the creep-fatigue life was applied. The founding is interesting and the prediction of creep-fatigue life is also very important for engineering materials.
